# Nutritional Characterization of Chilean Landraces of Common Bean

**DOI:** 10.3390/plants13060817

**Published:** 2024-03-12

**Authors:** Katherine Márquez, Osvin Arriagada, Ricardo Pérez-Díaz, Ricardo A. Cabeza, Andrea Plaza, Bárbara Arévalo, Lee A. Meisel, Daniela Ojeda, Herman Silva, Andrés R. Schwember, Camila Fuentes, Mónica Flores, Basilio Carrasco

**Affiliations:** 1Centro de Estudios en Alimentos Procesados (CEAP), Campus Lircay, Talca 3480094, Chile; arriagada.lagos.o@gmail.com (O.A.); jrperezd@gmail.com (R.P.-D.); aplaza@ceap.cl (A.P.); barevalo@ceap.cl (B.A.); cmla.fuentes@gmail.com (C.F.); 2Departamento de Producción Agrícola, Facultad de Ciencias Agrarias, Universidad de Talca, Talca 3460000, Chile; rcabeza@utalca.cl; 3Laboratorio de Genética Molecular Vegetal, Instituto de Nutrición y Tecnología de los Alimentos, Universidad de Chile, Santiago 7830490, Chile; lmeisel@inta.uchile.cl (L.A.M.); daniela.ojeda@inta.uchile.cl (D.O.); 4Laboratorio de Genómica Funcional & Bioinformática, Facultad de Ciencias Agronómicas, Universidad de Chile, Santiago 8820808, Chile; hesilva@uchile.cl (H.S.); monicafloresr@ug.uchile.cl (M.F.); 5Departamento de Ciencias Vegetales, Facultad de Agronomía e Ingeniería Forestal, Pontificia Universidad Católica de Chile, Santiago 7820436, Chile; aschwember@uc.cl

**Keywords:** amino acids, anti-nutritional, Chilean landrace, mineral content, *Phaseolus vulgaris*, protein

## Abstract

Common bean (*Phaseolus vulgaris* L.) is the primary grain legume cultivated worldwide for direct human consumption due to the high nutritional value of its seeds and pods. The high protein content of common beans highlights it as the most promising source of plant-based protein for the food industry. Additionally, landraces of common bean have great variability in nutritional traits, which is necessary to increase the nutritional quality of elite varieties. Therefore, the main objective of this study was to nutritionally characterize 23 Chilean landraces and 5 commercial varieties of common bean to identify genotypes with high nutritional value that are promising for the food industry and for genetic improvement programs. The landrace Phv23 (‘Palo’) was the most outstanding with high concentrations of minerals such as P (7.53 g/kg), K (19.8 g/kg), Mg (2.43 g/kg), Zn (52.67 mg/kg), and Cu (13.67 mg/kg); essential amino acids (364.8 mg/g protein); and total proteins (30.35 g/100 g seed). Additionally, the landraces Phv9 (‘Cimarrón’), Phv17 (‘Juanita’), Phv3 (‘Araucano’), Phv8 (‘Cabrita/Señorita’), and Phv4 (‘Arroz’) had a high protein content. The landrace Phv24 (‘Peumo’) stood out for its phenolic compounds (TPC = 218.1 mg GA/100 g seed) and antioxidant activity (ORAC = 22,167.9 μmol eq trolox/100 g extract), but it has moderate to low mineral and protein concentrations. In general, the concentration of nutritional compounds in some Chilean landraces was significantly different from the commercial varieties, highlighting their high nutritional value and their potential use for the food industry and for genetic improvement purposes.

## 1. Introduction

Common bean (*Phaseolus vulgaris* L.) is one of the most cultivated and consumed legumes in the world, especially in Latin America and Africa [1,2]. Common bean is the most important species among the genus *Phaseolus* L. cultivated for direct human consumption, contributing about 30% of the total daily protein intake in developing countries [3]. In addition, the high protein content of common beans is associated with increased satiety, and hence its consumption could help to treat obesity [4]. This legume is mainly consumed as dry beans or fresh vegetable in the form of snap bean pods [2]. Due to its high nutritional value and the demand for healthier foods [5], the world production of common beans reached 28.3 million metric tons in 2022, cultivated on a global area of about 36.7 million hectares, where the main producing region was Asia, followed by America and Africa [6]. 

Nutritionally, common beans are recognized as a good and inexpensive source of proteins, which have a concentration 2 to 3 times higher than cereal grains [7]. In fact, bean grains have between 20 and 30% protein; contain essential amino acids, folates, and complex carbohydrates such as dietary fiber; and are low in saturated fats [7,8]. They are also a rich source of vitamins such as folates, tocopherols, thiamine, riboflavin, niacin, biotin, and pyridoxine, and minerals such as potassium, phosphorus, calcium, iron, and zinc [1,7]. Additionally, common beans are characterized by a high content of bioactive compounds (flavonoids, tannins, phenolic, and other antioxidants), which have properties to reduce the risk of cancer and other chronic diseases [9,10]. Despite their nutritional benefits, common beans also contain anti-nutrients such as some enzyme inhibitors (α-amylase, trypsin, chymotrypsin), phytic acid, flatulence factors (oligosaccharides), and lectins, which affect the bioavailability and digestibility of nutrients and minerals [11]. 

Both nutritional and anti-nutritional compounds are influenced by agronomic management [12], environmental conditions [13], and genotypes [14,15]. Traditionally, the main objective of phenotypic selection in common bean has been to increase the yield during the domestication and post-domestication processes [16]. Therefore, most modern cultivars have lower levels of nutrients and vitamins [17]. In this sense, exploring the diversity related to nutritional and anti-nutritional parameters in ancient, landrace, or wild common beans is a key prerequisite to increase the nutritional quality of modern varieties. The common bean is divided into two gene pools, Mesoamerican and Andean, which are eco-geographically, morphologically, and genetically different [18]. Within the Andean genetic pool, the Chilean race of common bean is particularly interesting as modern Chilean landraces maintain levels of variability and genetic identity like the ancestral common bean from Argentina, being recognized as a genetic reservoir of the current Andean gene pool [19] and therefore as a possible source of alleles associated with nutritional quality. In this context, the Chilean landraces of common bean represent an interesting genetic resource to evaluate their nutritional value for food and genetic improvement purposes. 

From a nutritional point of view, Chilean landraces of common bean have been poorly studied. In fact, the only study evaluating the nutritional composition was carried out by Paredes et al. [20], who found a wide variability of macro- and micro-elements such as N, Fe, and Zn in the seeds. Considering that landraces are significant sources of protein, carbohydrates, vitamins, and minerals which are necessary for food security and healthy food supply [21], the main goal of this research was to nutritionally characterize Chilean landraces of common bean to identify some promising landraces with high nutritional value for direct consumption, the development of functional ingredients, and breeding purposes. 

## 2. Results

### 2.1. Variability among Chilean Common Beans

A summary of the descriptive statistics (mean, standard deviation, and range) for the 31 nutritional traits evaluated in this study is shown in Table 1. Additionally, the raw data (Appendix A) and descriptive statistics for the 23 Chilean landraces and 5 commercial varieties of common beans are shown in Appendix A. The analysis of variance (ANOVA) indicated that there are significant differences between the common bean accessions for all the traits evaluated (Table 1; Appendix A), which will be discussed in the following sections.

### 2.2. Mineral Contents

In general, Fe (71.76 mg/kg) was the micro-element most abundant within the accessions evaluated, while Cu (10.25 mg/kg) was the one with the lowest concentration. For the macro-elements, K and Ca were the elements with the highest (16.13 g/kg) and lowest (1.64 g/kg) concentration, respectively (Table 1). The mineralogical analysis of the Chilean common bean accessions for macro- and micro-elements showed wide and significant variation for all minerals analyzed (Figure 1).

The Phv23 (‘Palo’) and Phv18 (‘Manteca’) Chilean landraces were those with the highest (31.5 g/kg) and lowest (21.1 g/kg) concentration of macro-elements, respectively (Figure 1A). Specifically, the Phv23 landrace was the one that presented the highest concentration for all macro-elements (P = 7.53 g/kg; K = 19.8 g/kg; Mg = 2.43 g/kg) except for Ca, where the Phv4 (‘Arroz’) landrace had the highest concentration (2.93 g/kg). The lowest concentrations of P (3.43 g/kg), K (13.43 g/kg), Ca (0.87 g/kg), and Mg (1.73 g/kg) were found for the Phv18 (‘Manteca’), Phv6 (‘Azufrado’), Phv8 (‘Cabrita/Señorita’), and Phv15 (‘Hallado Alemán’) Chilean landraces, respectively. For total micro-elements, the Phv19 (‘Mantequilla’) Chilean landrace was the one with the highest concentration (190.0 mg/kg), while the Phv24 (‘Peumo’) landrace was the one with the lowest (110.33 mg/kg) concentration (Figure 1B). Interestingly, the Phv23 (‘Palo’) landrace was the one with the highest concentration of Zn (52.67 mg/kg) and Cu (13.67 mg/kg). In fact, Phv23 (‘Palo’) had a high concentration of total micro-elements, which is significantly different with respect to commercial varieties. The commercial varieties presented intermediate values for all macro- and micro-elements, except for Mn, where the Phv20 (‘Miami’) variety had the highest concentration (19.0 mg/kg). 

In summary, for the total mineral concentration, the Chilean landrace Phv23 (‘Palo’) stands out among all the Chilean landraces evaluated, including the commercial varieties. The mean and standard deviation of all macro- and micro-elements for each accession are shown in Figure 2 and Figure 3, respectively.

The µXRF analysis showed that the highest Ca concentration was found in the coat seed, while K was mainly stored in the cotyledons (Figure 4A). The highest S concentration was found in the cotyledons while P was evenly distributed in both tissues (Figure 4B).

### 2.3. Amino Acid Profile and Protein Content

Glutamic acid (Glu) was the most abundant amino acid (120.06 mg/g of protein) in the 28 common bean accessions, while Trans-4-hidroxyproline (4Hyp) was the least abundant (1.79 mg/g of protein; Appendix A). The total concentration of essential amino acids (EAAs) showed a wide variation among the beans, where the Phv15 (‘Hallado Alemán’) and Phv28 (‘Tórtola’) had the highest (367.9 mg/g protein) and lowest (293.2 mg/g protein) concentration, respectively (Figure 5A). The Phv23 (‘Palo’) landrace, which had a high concentration of mineral elements, had the fifth highest concentration of EAAs; however, its concentration was not statistically different from the landrace with the highest concentration (Phv15: ‘Hallado Alemán’). In addition, Phv23 (‘Peumo’) was the one with the highest concentration of phenylalanine (59.4 mg/g protein), and also had high values of methionine (10.3 mg/g protein) and threonine (40.7 mg/g protein). The commercial varieties Phv21 (‘Negro’), Phv25 (‘Rojo Canda’), and Phv1 (‘Alubia’) had a high (366.6 mg/g protein), medium (342.4 mg/g protein), and low (311.5 mg/g protein) concentration of EAAs. The tryptophan (trp) concentration varied from 24.2 to 396.9 for Phv19 (‘Mantequilla’) and Phv26 (‘Sapito’) in the Chilean landraces, respectively (Appendix A). The total concentration of non-essential amino acids (NEAAs) varied significantly (Figure 5B), where the Chilean landraces Phv3 (‘Araucano’) and Phv28 (‘Tórtola’) had the highest (452.1 mg/g protein) and lowest (369.5 mg/g protein) concentration. 

The Chilean landraces Phv15 (‘Hallado Alemán’), Phv3 (‘Araucano’), and Phv23 (‘Palo’) were the most interesting for total amino acid concentration. A complete summary of the EAA and NEEA contents for all 28 common bean accession is shown in Appendix A, respectively.

The soluble and total protein concentration varied significantly among the different common bean accessions (Figure 6). The total protein concentration of Chilean landraces ranged from 20.31% (Phv19: ‘Mantequilla’) to 30.35% (Phv23: ‘Palo’), while the total protein concentration of the commercial varieties ranged from 21.27% (Phv25: ‘Rojo Canada’) to 24.33% (Phv7: ‘Blanco Español’). The high concentration of total proteins in the landrace Phv23 (‘Palo’) is in accordance with the high concentration of total amino acids. Interestingly, the six accessions with the highest total protein concentration correspond to the Chilean landraces Phv23 (‘Palo’), Phv9 (‘Cimarrón’), Phv17 (‘Juanita’), Phv3 (‘Araucano’), Phv8 (‘Cabrita/Señorita’), and Phv4 (‘Arroz’), which were statistically different from the commercial varieties (Figure 6A). 

The concentration of soluble proteins (SPs) varied widely among the Chilean landraces (Figure 6B). Phv24 (‘Peumo’) and Phv11 (‘Coscorrón’) had the highest (10.01 mg/mL) and lowest (1.77 mg/mL) SP concentration, respectively. The Phv24 (‘Peumo’), with high SPs, had a moderate concentration of total protein. On the contrary, the Phv23 (‘Palo’), with high TP, also had a moderate concentration of soluble proteins, indicating a possible negative correlation between these two (SP and TP) traits. 

### 2.4. Sucrose and Raffinose Content

The raffinose (RFO) concentration of the Chilean common beans ranged significantly from 5.4 mmol/100 g in Phv3 (‘Araucano’) and Phv26 (‘Sapito’) to 3.92 mmol/100 g in Phv24 (‘Peumo’). However, the RFO concentration in Pvh24 (‘Peumo’) was not significantly different from the commercial varieties Phv25 (‘Rojo Canada’), Phv21 (‘Negro’), and Phv20 (‘Miami’), which had concentrations of 4.61, 4.58, and 4.16 mmol/100 g (Figure 7). Among the commercial varieties, the RFO concentration ranged from 4.16 mmol/100 g in Phv20 (‘Miami’) to 5.34 mmol/100 g in Phv1 (‘Alubia’) and Phv7 (‘Blanco Español’). In addition, the sucrose concentration exhibited greater variation than raffinose (Appendix A). In the Chilean landraces, the sucrose concentration ranged from 1.1% to 4.75% in Phv6 (‘Azufrado’) and Phv18 (‘Manteca’), respectively. The commercial varieties presented moderate concentration of sucrose varying from 2.25% to 3.47% in the varieties Phv20 (‘Miami’) and Phv21 (‘Negro’), respectively. 

### 2.5. Antioxidant Activity and Phenolic Concentration

The total phenolic concentration (TPC) and oxygen radical absorbance capacity (ORAC) varied widely among the 28 common bean accessions (Figure 8). The Chilean landraces Phv24 (‘Peumo’) and Phv11 (‘Coscorrón’) had the highest (218.1 mg GA/100 g) and lowest (49.1 mg GA/100 g) TPC, while the commercial varieties had moderate to low TPC (Figure 8A). The ORAC varied from 4544.9 to 23,045.0 umol eq trolox/100g extract in the Chilean landraces Phv11 (‘Coscorrón’) and Phv3 (‘Araucano’), respectively (Figure 8B). The commercial variety Phv20 (‘Miami’) stands out for having a high ORAC (22,366.86 umol eq trolox/100 g extract) and TPC (150.30 mg GA/100 g). Additionally, the landrace Pvh24 (‘Peumo’), which has the highest TPC and one of the highest ORACs, had the lowest raffinose concentration and a moderate sucrose concentration, which suggests that Phv24 (‘Peumo´) can be used for direct consumption or the development of functional ingredients.

### 2.6. PCA and Correlations between the Traits 

The principal component analysis based on sucrose, TPC, ORAC, total proteins (TP), soluble proteins (SP), raffinose (RFO), minerals (macro and micro), and amino acids (essential and non-essential) is shown in Figure 9A. Principal component 1 (PC1) and PC2 explained 37.3% and 19.2% of the total variance, respectively. There was no clear differentiation between the Chilean landraces and the commercial varieties (Figure 9B). PC1 was positively correlated with EAAs, NEAAs, ORAC, TPC, and SP, while it was negatively correlated with micro-elements. The loading plot from the PCA analysis indicated that SP and TPC contributed the most to the differentiation between accessions (Appendix A). 

PCA based on minerals (micro and macro-elements) could not differentiate between the Chilean landraces and commercial varieties either (Appendix A), where PC1 and PC2 explained 35.2% and 25.1% of the total variation, respectively. PC1 was negatively correlated with K, P, Mg, and Cu concentrations, while PC2 was positively correlated with B and negatively correlated with Ca (Appendix A). In addition, the PCA based on amino acid (EAA and NEAA) concentration no differentiate between the Chilean landraces and the commercial varieties (Appendix A), where PC1 and PC2 explained 46.8% and 15.5% of the total variation, respectively. PC1 was positively correlated with Leu, Ile, and Val, and negatively correlated with Trp, while PC2 was positively correlated with Asp and 4Hyp (Appendix A). 

Finally, the correlations between all the evaluated nutritional and anti-nutritional traits are shown in Figure 10. In general, most amino acids showed a significant and positive correlation between them, except for tryptophan and alanine which showed a negative correlation. A moderate positive correlation (*r*^2^ = 0.38*) exists between total protein concentration (nutritional compound) and RFO concentration (anti-nutritional compound). Additionally, soluble proteins (SPs) had a high and positive correlation with compounds related to antioxidant capacity such as TPC (*r*^2^ = 0.89*) and ORAC (*r*^2^ = 0.81*). In contrast, TPC had moderate negative correlations with some minerals such as Zn (*r*^2^ = −0.38), Fe (*r*^2^ = −0.41), and Ca (*r*^2^ = −0.39), while ORAC had a negative correlation with Fe (*r*^2^ = −0.38). 

## 3. Discussion

### 3.1. Variability among Chilean Common Beans

Significant variability was found for the 31 nutritional and anti-nutritional traits in the 23 Chilean landraces and 5 commercial varieties of common bean evaluated. Considering that the Chilean landraces mainly have been evaluated from a genetic point of view [19,22,23], and are poorly studied nutritionally [20], these results are of great value since they confirm the potential of Chilean landraces to be used in the processed food industry, especially to make healthy food products as well as for the genetic improvement of the nutritional quality of commercial varieties.

### 3.2. Mineral Contents 

The only study evaluating nutritional compounds in Chilean landraces was performed by Paredes et al. [20], who reported a wide variability in macro- and micro-elements. For macro-elements, P ranged from 4.0 to 5.6 g/kg, K varied between 14.2 and 18.4 g/kg, Ca ranged from 1.0 and 2.6 g/kg, and Mg ranged from 1.3 and 2.3 g/kg [20], which were like those reported in this study. Furthermore, the concentration of these macro-elements in the Chilean landraces was like those reported for 25 American and European common bean accessions, belonging mainly to the Mesoamerican genetic pool [24]. On the contrary, a higher concentration of Ca (5.86 to 12.29 g/kg) and a lower concentration of Mg (0.64–1.10 mg/kg) were found in 29 common bean genotypes from CIAT and USA including both gene pools [25]. 

From a nutritional standpoint, P concentration is important in common bean seeds because it is related to phytate concentration [26]. In fact, phytic acid is recognized as an efficient chelator of nutritionally important mineral cations; therefore, it is an anti-nutrient for different essential minerals in the diet, especially Fe and Zn [27]. The Chilean landraces stand out for their high Fe (49–109 mg/kg) and Zn (26–48 mg/kg) concentration in comparison with Brazilian varieties of *P. vulgaris* whose Fe and Zn concentration varied from 14.4 to 27.3 mg/kg and 12.1 to 26.7 mg/kg, respectively [28]. Additionally, in a core collection from the International Center for Tropical Agriculture (CIAT) that includes about 1100 genotypes from both gene pools, the Fe concentration ranged from 35 mg/kg to 92 mg/kg, with a mean of 55 mg/kg [29]. However, the Fe and Zn concentrations were slightly lower compared to accessions from the Mesoamerican genetic pool [24]. A negative correlation has been evidenced between the phytate content and the content of Fe and Zn [28]. Given that in our study high contents of Fe and Zn were found in the Chilean landraces, a low content of phytates could be expected. 

Among the Chilean landraces, Phv23 (‘Palo’) is highlighted due to its high concentration of micro- (Cu, Zn, Mn) and macro-elements (Mg, P, and K) that are key for human health. Therefore, this landrace could be a good candidate to increase the content and bioavailability of some important minerals in commercial varieties to improve mineral intake of the population.

### 3.3. Amino Acid Profile and Protein Content

One of the objectives of this study was to identify Chilean landraces of common bean with potential uses in the food industry as raw materials for plant-based products, and one of the most important traits for this purpose is the protein concentration of the seeds. The average total protein (TP) was 24.19% for the 28 accessions of common beans and 24.58% for the 23 Chilean landraces. The TP for Chilean landraces was higher than other races of *P. vulgaris* from the Mesoamerica gene pool, whose TP ranged from 19.3 to 23.03% [28,30]. The landraces Phv23 (‘Palo’), Phv9 (‘Cimarrón’), Phv17 (‘Juanita’), Phv3 (‘Araucano’), Phv8 (‘Cabrita/Señorita’), and Phv4 (‘Arroz’) have a high protein concentration and are statistically different from commercial varieties, suggesting that they could be used as raw materials for the development of plant-based products, thus reflecting the high nutritional value of these Chilean landraces. Additionally, these landraces can be useful in breeding programs to increase the protein concentration of available commercial varieties. 

The main essential amino acids found in common beans were leucine and lysine, while sulfur amino acids such as methionine were present in low concentrations. On the other hand, cysteine was not detectable in this study. However, it has been reported that *P. vulgaris* accumulates large amounts of non-protein sulfur amino acid derivatives, such as S methyl Cys and γ glutamyl S methyl Cys, in mature seeds [31]. Considering that common beans contain low levels of sulfur amino acids such as methionine and cysteine [32], the Chilean landraces Phv15 (‘Hallado Alemán’) and Phv3 (‘Araucano’) have an excellent potential to be used in the food industry due to their remarkable levels of methionine. On the other hand, tryptophan (Trp) is one of the most oxidizable amino acids and it has been proposed to be relevant for human nutrition because it is essential in the synthesis of melatonin and serotonin, as well as vitamin B3 [33]. A high signal was found for Phv26 (‘Sapito’), indicating that this typical Chilean landrace has a high content of this essential amino acid, which has been discussed as a potential protective factor for physical and mental health [34].

### 3.4. Sucrose and Raffinose Content

A high raffinose concentration in legumes may limit their consumption due to the production of gases and digestive disturbances [35]. However, raffinose play an essential role in plant biology by protecting embryos during desiccation and serves as an energy source during germination [36]. Furthermore, raffinose also protects plants against different types of abiotic stress such as heat, drought, salt, and oxidative stresses [37]. Although sucrose and myo-inositol are precursors of raffinose in legume seeds and affect their accumulation at maturity [38], there was no significant correlation between RFO and sucrose accumulation in mature seeds in this study. In contrast, other legumes such as chickpea have showed a positive correlation between raffinose and myo-inositol and sucrose, especially during the early seed developmental stages [39]. The Chilean landrace Phv24 (‘Peumo’) has the potential to be used in genetic improvement programs to reduce the raffinose in seeds. 

### 3.5. Antioxidant Activity and Phenolic Content

Evidence from medical studies indicates that the consumption of polyphenol-rich foods can help mitigate cardiovascular and non-communicable diseases such as diabetes and cancer [40,41,42]. The Chilean landraces evaluated have a high concentration of phenols ranging from 49.18 mg GAE/100 g (Phv11: ‘Coscorrón’) to 218.11 mg GAE/100 g (Phv24: ‘Peumo’), with an average of 112.76 mg GAE/100 g. Some authors indicate that the phenolic concentration of the Mesoamerican and the Andean gene pools are in the range of 35 to 389 mg GAE/100 g [43]. Similarly, Espinosa-Alonso et al. [44] reported a TPC range between 90 and 211 mg GAE/100 g for the genotypes of the Durango and Jalisco races. In this sense, the content of phenolic compounds in Chilean landraces is within the values obtained for the genotypes of both genetic pools (Mesoamerican and Andes). However, Valdés et al. [45] reported a phenolic concentration between 350 and 500 mg GAE/100 g for genotypes from Brazil. 

According to the content of total phenolic compounds (TPCs) and antioxidant activity (ORAC), the landrace Phv24 (‘Peumo’) is one of the most promising to increase the phenolic compounds and antioxidant activity in bean varieties for health purposes. 

## 4. Materials and Methods

### 4.1. Plant Materials

A total of 28 common bean accessions were evaluated for their nutritional composition in this study: 23 correspond to Chilean landraces and 5 correspond to commercial varieties cultivated in Chile. These accessions have different growth habits varying from type I, characterized by plants that present terminal reproductive buds and inflorescences (determinate), to type IV, characterized by tall indeterminate plants with long vines, terminal vegetative buds, and strong climbing ability [46], including the intermediate growth habits II and III (Figure 11; Appendix A). Chilean landraces were collected from local farmers in the central–south regions of the country (O’Higgins to Maule regions). A field experiment was conducted in Curepto (35°05′ S; 72°01′ W; 9 m.a.s.l), a coastal area of the Maule region. The common bean genotypes were arranged in a randomized complete block design with three replicates. Plots were arranged as single rows with 3 plants per plot.

### 4.2. Mineral Composition

The seeds were oven-dried at 65 °C, finely milled, and then calcinated at 550 °C, and the ashes were subsequently digested with HCl 2N. Flame atomic absorption spectrometry (FAAS) (Agilent Technologies 280 FS/240 FS, Santa Clara, CA, USA) was used to determine the concentration of K, Ca, Mg, Cu, Zn, Mn, and Fe. Total N was determined by an elemental analyzer (TruSpec CN Leco^®^, St. Joseph, MI, USA). P and B concentrations were measured by colorimetry (Agilent Cary 8454 UV/Vis, Santa Clara, CA, USA) using the vanadate–molybdate method and the azomethine-H method, respectively. The chemical analyses were performed according to the standard methodologies used in Chile, which were published by Sadzawka et al. [47]. Total protein (TP) content was calculated by the factor of 6.25 according to the Dumas method [46]. Elemental mapping of common bean seeds was obtained using micro X-ray fluorescence (μ XRF) spectroscopy (BrukerTM M4 Tornado PlusTM, Bremen, Germany). For the measurements, the X-ray was operated at 50 kV and 30 μA (Rh anode with a Kα = 20.6), with a spot size of 20 μm, and an energy-dispersive silicon drift detector (30 mm^2^ and 142 eV for Mn Kα, XFlashTM, Singapore). Measurements were taken under 2 mbar vacuum conditions; the images were produced using a 20 μm × 20 μm pixel and an X-ray exposition time of 20 ms per pixel.

### 4.3. Amino Acids Composition

Amino acid hydrolysis was carried out according to AOAC 994.12 [48] to determine all essential amino acids (EAAs: glycine, histidine, threonine, valine, methionine, isoleucine, leucine, phenylalanine, and lysine) and non-essential amino acids (NEAAs: aspartic acid, glutamic acid, trans-4-hydroxyproline, serine, arginine, alanine, proline, and tyrosine), except for tryptophan and cysteine. Briefly, seeds were milled finely and subsequently hydrolyzed in 6 N HCl. Samples were derivatized, and individual amino acids were separated by HPLC using an Inerstil C18 ODS-3 250 mm × 5 µm × 4.6 mm column (GL Science, Dillenburgstraat, Netherlands) according to the literature [49]. Amino acid standard mix (0.015 to 0.240 mg/mL) calibration curves were used to identify and quantify the amino acids.

### 4.4. Sugars Content

Raffinose (RFO) was quantified spectrophotometrically at 510 nm (Tecan Infinite M200 PRO-Nano Quant, Mannedorf, Switzerland) using the Megazyme Raffinose/Sucrose Glucose Assay Kit according to the manufacturer’s recommendations (Megazyme^®^, Bray, Ireland). The determination of sugars (glucose, fructose, maltose, lactose, and sucrose) was performed by HPLC (Agilent 1200 series, Santa Clara, CA, USA) with a refraction index (RI) detector. For the pre-treatment of samples, 9 mL of 50% ethanol was added to a 1.0 g common bean sample, shaken in an orbital shaker for 1 h, and then centrifuged for 15 min at 5000 rpm, and the supernatant was subsequently injected into an ultra-amino column (4.6 × 150 mm, 3 µm particle; Restek, Bellefonte, PA, USA) set at 40 °C, and then the RI detector at 32 °C. The mobile phase was acetonitrile/water 85/15% *v*/*v* at 1.2 mL/min for 35 min. The respective calibration curves were prepared with a mix of sugar standards with a concentration between 5 and 75 g/L [50].

### 4.5. Total Phenolic Content (TPC)

TPC was determined by the Folin–Ciocalteu method [51]. For the pretreatment of samples, 8 mL of 60% ethanol was added to 2.0 g of bean flour, shaken in an orbital shaker for 1 h (400 rpm), and then centrifuged for 15 min at 6000 rpm, and the supernatant was subsequently measured. Gallic acid was employed for the calibration curve. Briefly, in a transparent plate with 96 wells, 200 µL of Folin’s reagent was added to each well using 15 µL of the sample, 40 µL of sodium carbonate solution (7.5% *w*/*v*), and 45 µL of MilliQ water, then the transparent plate was placed in a microplate reader (Synergy HTX Multi-Mode Reader, Biotek, Winooski, VT, USA). Finally, absorbance was measured at 750 nm and the results are expressed as mg GAE (Gallic Acid Equivalent)/100 g of sample [52].

### 4.6. Antioxidant Activity

A microplate reader (Synergy HTX Multi-Mode Reader, Biotek, USA) was used to analyze the oxygen radical absorbance capacity (ORAC). The samples were prepared by weighing 0.5 g and adding 20 mL of the acetone/water extraction solvent (50 + 50, *v*/*v*). The sample was shaken at 400 rpm at room temperature in an orbital shaker for 1 h. Once the shaking was over, the extracts were centrifuged at 7000 rpm for 30 min, and the supernatant was ready for analysis after dilution with buffer solution. For the measurement, black plates with 96 wells were used. Briefly, sodium fluorescein (0.015 mg/mL), AAPH radical solution (120 mg/mL), and Trolox standard solution (100 µM) were prepared with phosphate buffer (75 mM, pH 7). The operating conditions for the final reaction consisted of 50 μL of diluted extract, Trolox standard or phosphate buffer (blank), 50 μL of fluorescein, and 25 μL of AAPH incubated at 37 °C in the microplate reader. The fluorescence was recorded every 5 min over 60 min, differences in the areas under the fluorescence decay curve (AUC) between the blank and the sample over time were compared, and the results were expressed as μM Trolox Equivalents [53].

### 4.7. Determination of Tryptophan and Its Oxidation Products

The soluble protein (SP) concentration was determined in microplates using the bicinchoninic acid assay (BCA protein assay, Thermo Scientific, Rockford, IL, USA). The calibration curves were prepared using bovine serum albumin (BSA) as a standard protein. The absorbance was measured at 562 nm in a microplate reader (ELx800, BioTek, Winooski, VT, USA). The same extracts described in the determination of TPC were used, previously concentrated by rapid vacuum (Thermo Scientific Savant SpeedVac SPD121P) and reconstituted in 200 μL of distilled water. Protein was adjusted to have a final concentration of 0.161 mg/mL, diluting with phosphate buffer (100 mM, pH 7.4) to a final volume of 1 mL. Steady-state fluorescence for tryptophan (Trp) and its oxidation products N Formylkynurenine (N formyl Kyn) and kynurenine (Kyn) was determined in a fluorescence spectrofluorometer (L.S. 55, Perkin Elmer, Waltham, MA, USA). The excitation/emission wavelengths were set to 295 nm/360 nm for Trp, 325 nm/434 nm for N-formyl Kyn, and 365 nm/480 nm for Kyn.

### 4.8. Statistical Analyses

All statistical analyses were performed using Rstudio 2023.09.1 software. Descriptive statistics for each nutritional trait were calculated in the package ‘dplryr’ using the summarize function [54]. Pearson’s correlation between all variables was implemented with the package ‘corrplot’ [55]. One-way analysis of variance (ANOVA) was performed using the ‘aov’ function to determine differences between the means of the common bean accessions for all the traits evaluated, except for tryptophan which was measured only once. Statistically significant differences (*p* < 0.05) between the means were determined by Tukey’s tests using the TukeyHSD function from the ‘agricolae’ package [56]. Three principal component analyses (PCAs) were performed and visualized using the ‘Factoextra’ package [57]. A PCA was performed using the following ten variables: total micro- and macro-elements, total essential (EAAs) and non-essential (NEAAs) amino acids, SP, ORAC, TPC, RFO, sucrose, and protein. Then, individual PCAs were performed for micro- and macro-elements and essential and non-essential amino acids. Finally, scatterplots and barplots were graphed using the ‘ggplot2′ package [58].

## 5. Conclusions

The present study allowed the identification of a group of Chilean landraces with high potential to be used in the food industry to develop functional foods, as well as to improve the nutritional value of commercial varieties through genetic improvement. However, the landraces highlighted in this study are mainly of growth habit III and IV. Therefore, they are useful and recommended for small farmers. 

## Figures and Tables

**Figure 1 plants-13-00817-f001:**
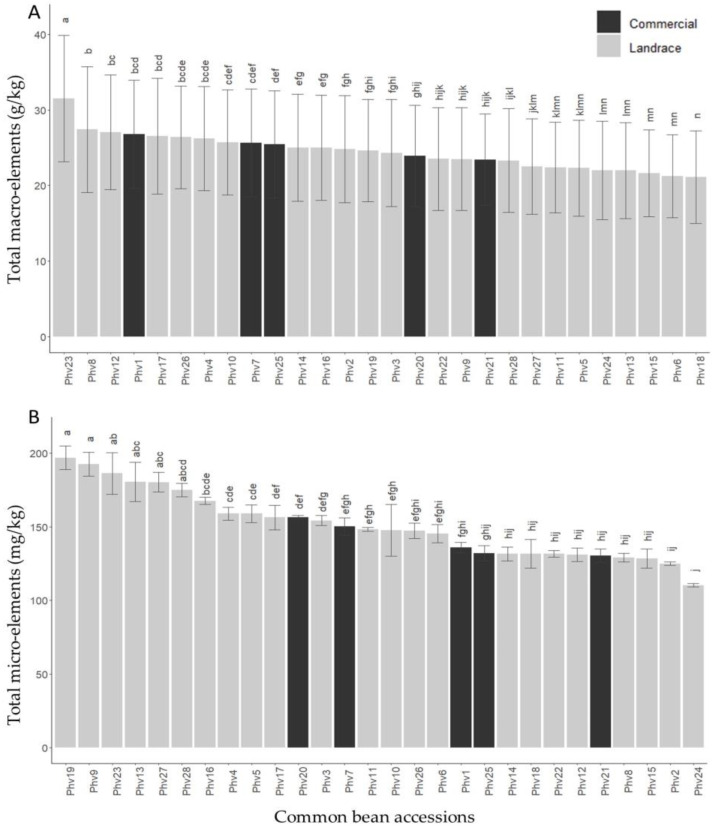
Total macro- (**A**) and micro-element (**B**) concentrations for the 28 common bean accessions evaluated. Bars with different letters are significantly different (*p* < 0.05) according to the Tukey test. The abbreviations Phv1 to Phv28 correspond to the 28 accessions of *Phaseolus vulgaris* (Phv) used in this study. For more details about the accessions, see Appendix A.

**Figure 2 plants-13-00817-f002:**
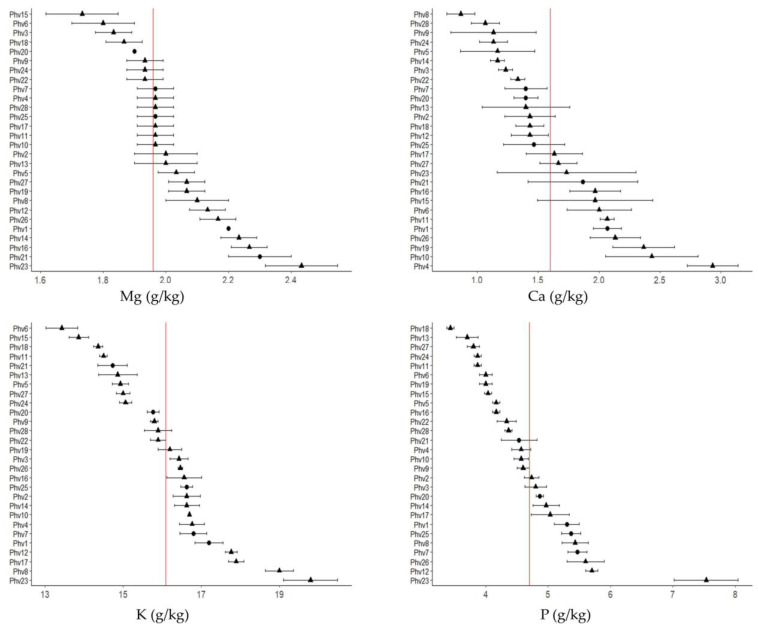
Mean concentration of macro-elements (Mg, Ca, K, and P) for the 28 common bean accessions. Chilean landraces and commercial varieties are indicated with a triangle and a circle in the scatterplots, respectively. The red vertical line indicates the average concentration of each element in all accessions. Significant differences between each accession according to the Tukey test are shown in Appendix A.

**Figure 3 plants-13-00817-f003:**
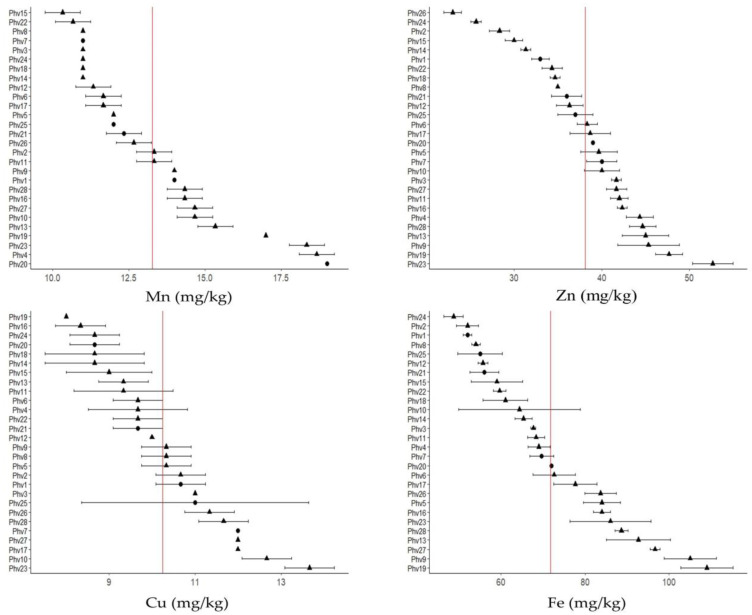
Mean concentration of micro-elements (Mn, Zn, Cu, and Fe) for the 28 common bean accessions. Chilean landraces and commercial varieties are indicated with a triangle and a circle in the scatterplots, respectively. The red vertical line indicates the average concentration of each element in all accessions. Significant differences between each accession according to the Tukey test are shown in Appendix A. The boron (B) concentration is shown in Appendix A.

**Figure 4 plants-13-00817-f004:**
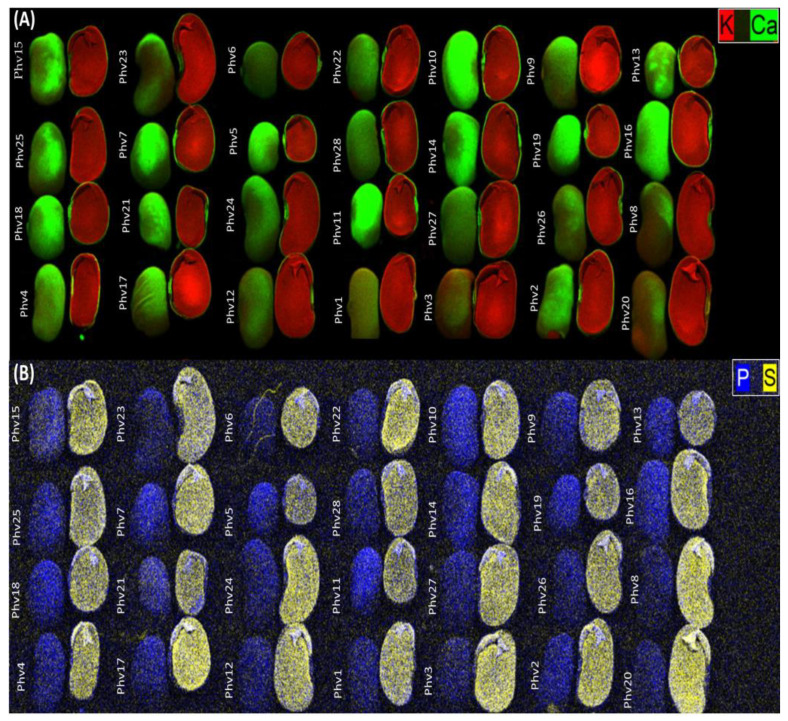
Micro X-ray (µXRF) fluorescence image of K and Ca (**A**), and P and S (**B**) distribution in seeds of 28 common bean accessions. K localization is indicated in red, Ca in green, P in blue, and S in yellow.

**Figure 5 plants-13-00817-f005:**
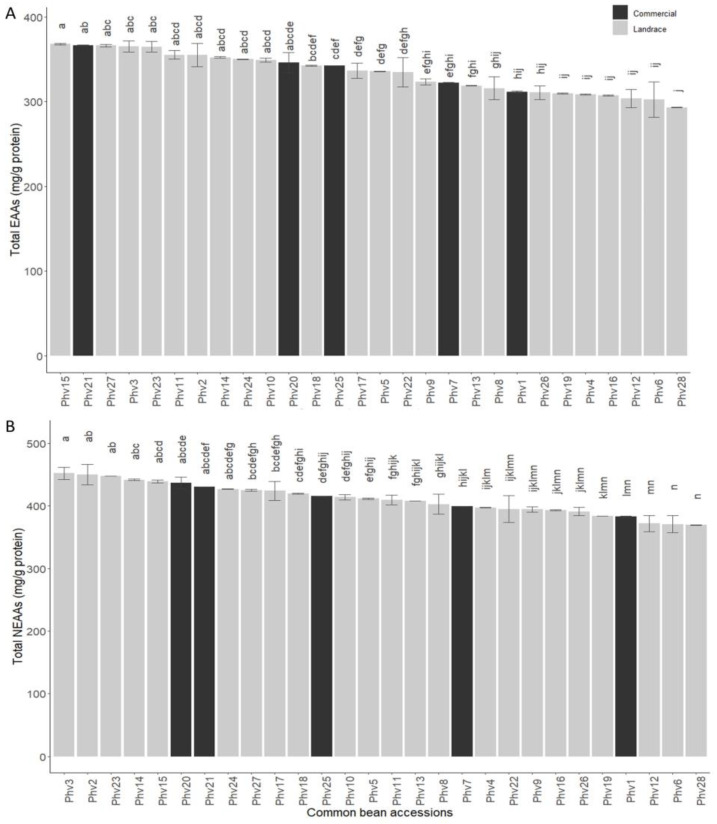
Total essential (**A**) and non-essential (**B**) amino acid contents for the 28 common bean accessions evaluated. Bars with different letters are significantly different (*p* < 0.05) according to the Tukey test.

**Figure 6 plants-13-00817-f006:**
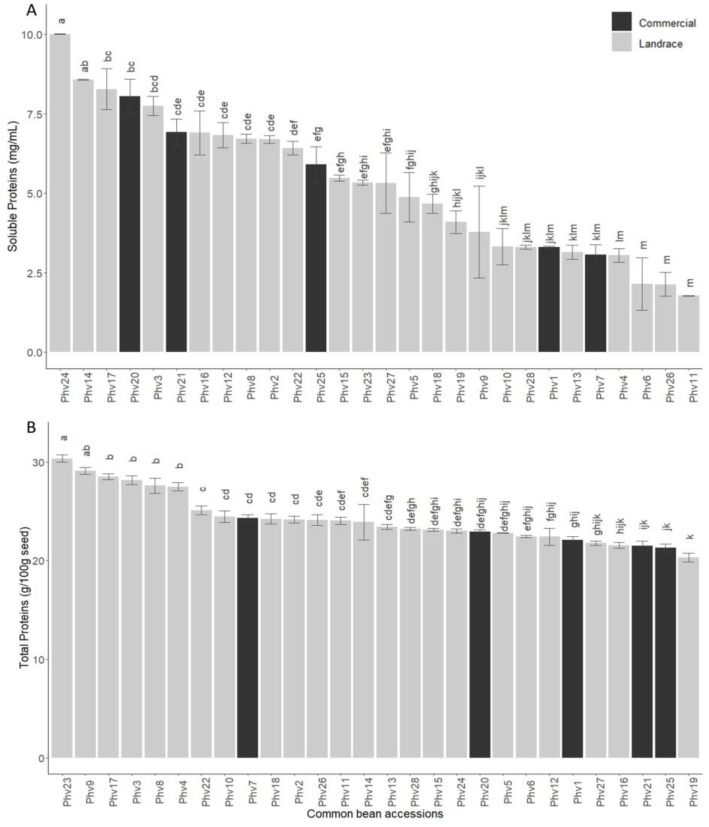
Soluble (**A**) and total (**B**) protein contents for the 28 common bean accessions evaluated. Bars with different letters are significantly different (*p* < 0.05) according to the Tukey test.

**Figure 7 plants-13-00817-f007:**
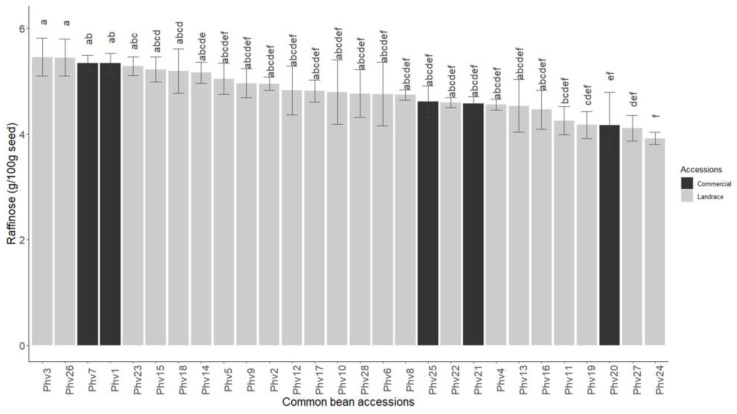
Raffinose (RFO) concentration for the 28 common bean accessions evaluated. Bars with different letters are significantly different (*p* < 0.05) according to Tukey test.

**Figure 8 plants-13-00817-f008:**
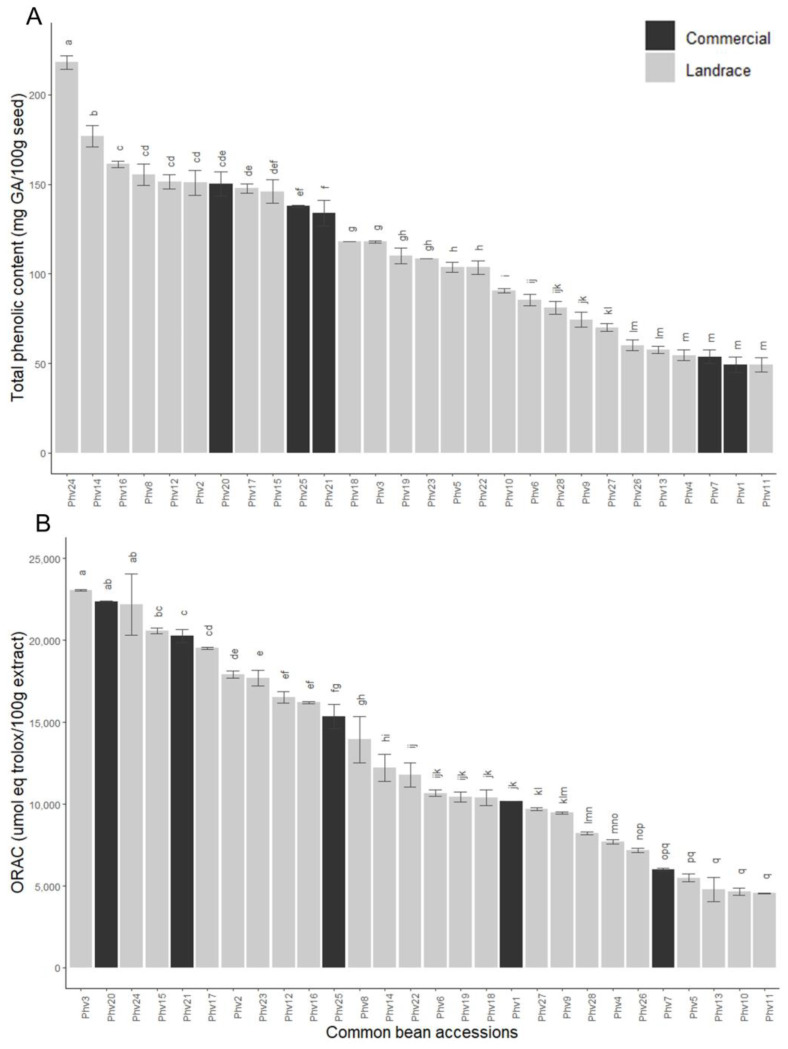
Total phenolic concentration (**A**) and ORAC (**B**) for the 28 common bean accessions evaluated. Bars with different letters are significantly different (*p* < 0.05) according to the Tukey test.

**Figure 9 plants-13-00817-f009:**
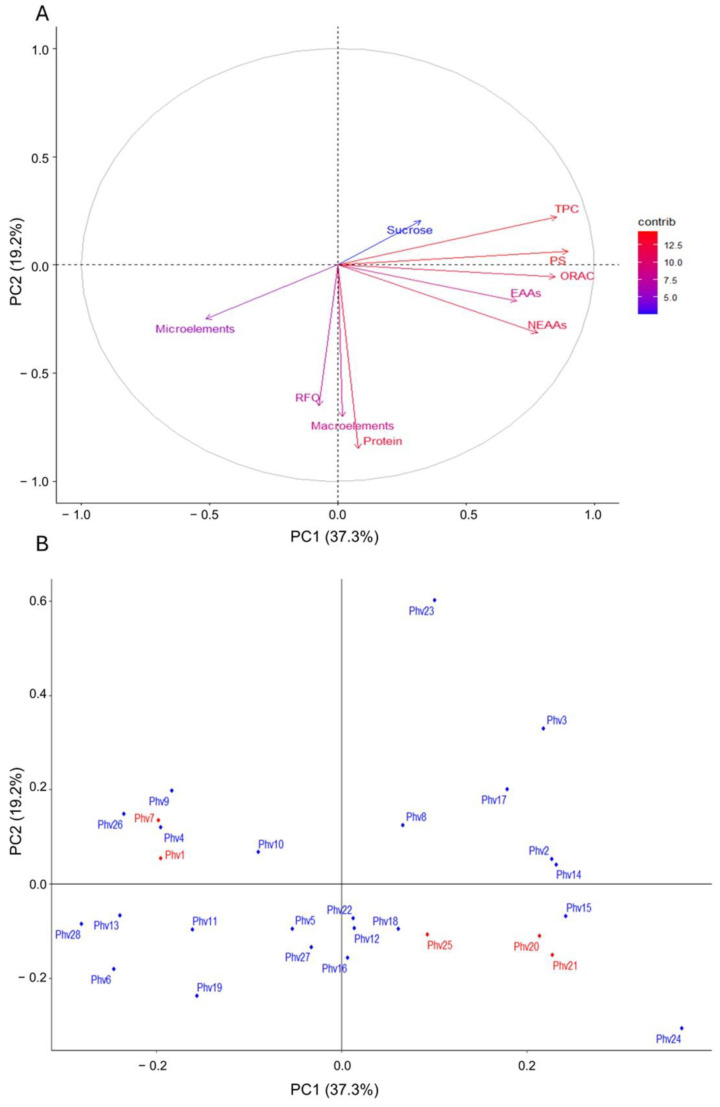
(**A**) Loading plot from the principal component analysis. (**B**) Principal component analysis for the 28 accessions based on sucrose, total phenolic concentration (TPC), ORAC, total protein (TP), soluble protein (SP), raffinose (RFO), minerals (macro and micro), and amino acids (essential and non-essential).

**Figure 10 plants-13-00817-f010:**
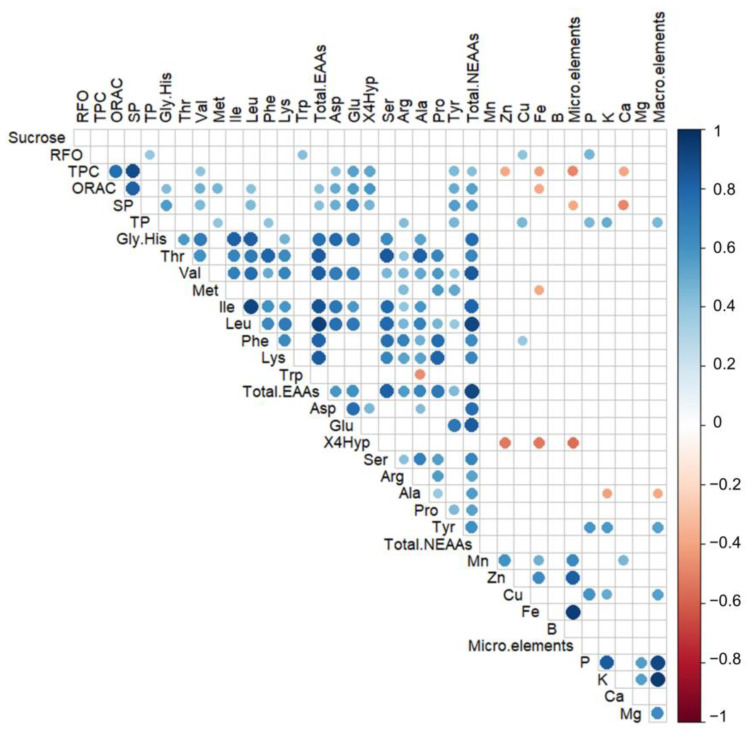
Correlation plot matrix between all traits for the 23 Chilean landraces and 5 commercial varieties of common bean. Only significant correlations are shown (*p* < 0.05). The size and color of the circles indicate the value of each correlation. For details of correlation values, see Appendix A.

**Figure 11 plants-13-00817-f011:**
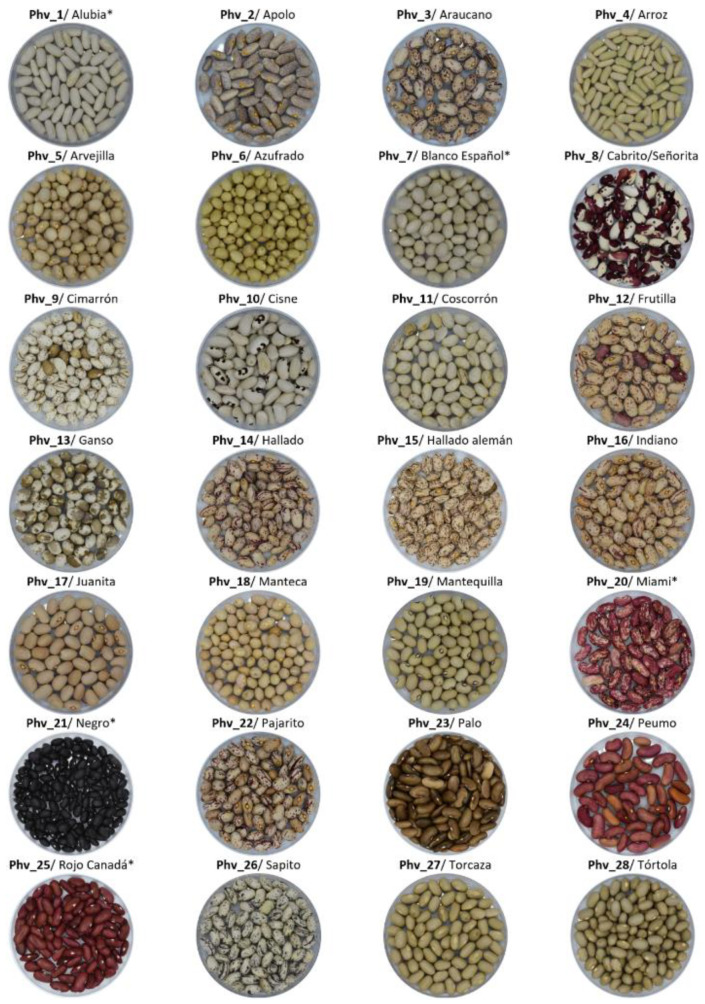
Shape and color of the 28 common beans used in this study. Phv1 (‘Alubia’), Phv7 (‘Blanco Español’), Phv20 (‘Miami’), Phv21 (‘Negro’), and Phv25 (‘Rojo Canada’) correspond to commercial varieties cultivated in Chile (*).

**Table 1 plants-13-00817-t001:** Mean, standard deviation (Sd), and range among the 28 common bean accessions for the nutritional quality traits analyzed.

Category	Trait	Unit	Mean	Sd	Min	Max	Significance
Sugars	Sucrose	g/100 g seed	2.61	0.85	1.11	4.75	*
	RFO	mmol/100 g seed	4.79	0.43	3.92	5.45	*
Antioxidant activity	TPC	mg GA/100 g seed	111.38	44.55	49.18	218.11	*
	ORAC	umol eq trolox/100 g extract	12,821.04	5936.16	4544.91	23,045.01	*
Protein content	SP	mg/mL	5.28	2.21	1.77	10.01	*
	TP	g/100 g seed	24.19	2.60	20.31	30.35	*
Macro-elements	P	g/kg	4.67	0.84	3.43	7.53	*
	K	g/kg	16.13	1.45	13.43	19.80	*
	Ca	g/kg	1.64	0.48	0.87	2.93	*
	Mg	g/kg	2.03	0.15	1.73	2.43	*
	Total	g/kg	24.47	2.31	21.10	31.50	*
Micro-elements	Mn	mg/kg	13.27	2.53	10.33	19.00	*
	Zn	mg/kg	38.13	6.71	23.00	52.67	*
	Cu	mg/kg	10.25	1.42	8.00	13.67	*
	Fe	mg/kg	71.76	16.66	48.67	109.00	*
	B	mg/kg	17.32	2.74	13.00	26.33	*
	Total	mg/kg	150.74	22.62	110.33	197.00	*
EAAs	Gly.His	mg/g protein	61.19	4.20	49.92	68.77	*
	Thr	mg/g protein	35.45	2.61	30.91	40.71	*
	Val	mg/g protein	39.59	2.79	34.48	45.72	*
	Met	mg/g protein	7.34	1.94	3.32	11.84	*
	Ile	mg/g protein	35.75	3.13	30.63	41.82	*
	Leu	mg/g protein	61.08	4.30	53.64	68.79	*
	Phe	mg/g protein	42.86	5.29	35.91	59.40	*
	Lys	mg/g protein	50.99	4.93	43.78	61.62	*
	Total	mg/g protein	334.24	23.23	293.27	367.94	*
NEAAs	Asp	mg/g protein	90.22	7.02	75.35	101.42	*
	Glu	mg/g protein	120.06	8.34	98.63	134.13	*
	4Hyp	mg/g protein	1.79	0.62	0.75	3.30	*
	Ser	mg/g protein	47.40	3.99	40.20	59.44	*
	Arg	mg/g protein	56.09	6.41	46.67	70.01	*
	Ala	mg/g protein	37.25	3.19	31.35	43.09	*
	Pro	mg/g protein	31.56	2.70	27.65	37.50	*
	Tyr	mg/g protein	26.33	4.11	18.76	34.36	*
	Total	mg/g protein	410.70	24.38	369.57	452.15	*

RFO: raffinose; TPC: total phenolic content; ORAC: antioxidant capacity; SP: soluble proteins; TP: total protein content; EAAs: essential amino acids; NEAAs: non-essential amino acids. * *p* < 0.05 according to ANOVA.

## Data Availability

Data are contained as Appendix A.

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
