# Peer review of "Nutritional Characterization of Chilean Landraces of Common Bean"

_plants, 2024, doi:10.3390/plants13060817_

Round 1

Reviewer 1 Report

Comments and Suggestions for Authors

Dear author(s),

Please find my comments and the suggested corrections. These should clearly be explained and addressed for readers:

Title

1.       Line (L) 2-3, the concise title will be better: “Nutritional characterization of Chilean common bean landraces”. Readers will read comments and explanations. 

Abstract

2.       L21, the aims of the study could be written.

3.       L22, change 5 to five.

4.       L25-26, the amount of mineral contents and protein could be removed from Abstract.

5.       L27-31, anti-nutritional, and phenolic contents should be removed. Readers can read them in the Results section.

Keywords

6.       L35, change Chilean landraces to landrace, remove the author name of the Phaseolus vulgaris, you can write “amino acids”, “phenolic contents”,  “sugar contents”

1. Introduction

7.       L40, write the author name of the genus as Phaseolus L.

8.       L44, consider to cite below article at the end of sentence: https://doi.org/10.3390/agriculture13050953

9.       L46, please update the data provided by FAOSTAT since 2022 data are available now.

2. Results

10.    L85, there were 31 traits in Table 1.

11.    L94, remove ANOVA

12.    L96, p should be italicized.

13.    L101, the subtitles should be the same. Change Contents to contents

14.    L131, p should be italicized.

15.    L142, change reported to found

16.    L243, check uppercase letters writing Trans-4-Hidroxyproline

17.    L291, p should be italicized.

18.    L294, change between to among

19.    L303-304, remove this sentence

20.    L345, p should be italicized.

21.    L364, p should be italicized.

22.    L434, p should be italicized.

23.    L469, change between to among

24.    L471-477, please put * instead of < 0.05. Asterisks will be better to read for readers.

25.    L511, p should be italicized.

3. Discussion

26.    L516, check 35 traits.

27.    L517, write five commercial cultivars

28.    L566, you can compare amino acids content with faba bean below study: https://doi.org/10.3390/agriculture13050932 

29.    Discussion could be enriched with the used and the suggested references

4. M&M

30.    L605, change Plant material to Plant materials

31.    L610, please write altitude

32.    L613, explain the growth habit of the geneotypes/accessions in Table S4 for readers.

33.    L642, please write the names of the amino acids.

34.    L683, change Bi- to bi-

35.    L686, change analysis to analyses

36.    L687, change analysis to analyses

37.    L703, p should be italicized.

38.    L708, change Sucrose to sucrose

39.    L708, change Protein to protein

5. Conclusion

40.    L716-719, remove the last sentence.

Tables

41.    In Tables, the captions should reflect the content of the study. Check whole Tables and correct Table S4.

 I hope you will find my suggestions useful.

Reviewer 2 Report

Comments and Suggestions for Authors

The paper “Nutritional characterization of Chilean landraces of common  bean: a valuable resource for the food industry and genetic improvement” by Katherine Márquez et al. reported data about the nutritional characterization of 23 Chilean bean landraces compared with 5 commercial varieties. Discussing the results obtained, the authors suggest the potential use of some of these varieties in food industries and for breeding purposes.

The results are interesting, but the presentation needs to be improved and the discussion needs to be more in-depth.

I suggest a careful revision of the manuscript; in different parts of the manuscript the sentences are not clear or contain typos, i.e. :

line 68-71: not clear “Within the Andean genetic 68 pool, the race Chile of common bean is particularly interesting due to modern Chilean landraces maintain levels of variability and genetic identity like ancestral common bean from Argentina, being recognized as a genetic reservoir of the current Andean gene pool [19]”.

Line 77: “…wide variability in seed element as….” I suggest adding “…wide variability in macro and micro element as...”

Table 1: I suggest adding the definition of the acronyms EAA and NEAA to the other definitions of the acronyms used in the table.

Figure 9: Fig 9A and 9B are too small, and the orange colour is too light and difficult to read

Line: 470-471 not clear: “Both essential and non-essential amino acids are positively correlated with each other.”

Line 517-519 not clear, this sentence appears to be incomplete

“Considering that the Chilean landraces has been evaluated mainly genetically [19,22,23], and poorly studied from a nutritional point of view [20].”

 Line 535-538: “From a nutritional standpoint, P concentration is important in common bean seeds 5 because it is related to phytate concentration [26], which is recognized as an antinutrient 5 for different essential minerals in the diet, especially Fe and Zn [27]. The Chilean landraces 535 stand out for their high Fe concentration (49 - 109 mg/kg) and Zn (26 - 48 mg/kg) in comparison with Brazilian varieties of P. vulgaris whose Fe and Zn concentration varied 14.4 537 to 27.3 mg/kg and 12.1 to 26.7 mg/kg, respectively [28].”

I suggest adding a comment to link the P content with the content of microelements, usually chelated by phytates, in the analyzed landraces.

Line 573: “In this regard, high signal was found  for Phv26 (´Sapito`) a typical Chilean landrace”.

I suggest adding a comment on the nutritional significance of this data.

Fig 10: I suggest moving figure 1 to the beginning of the article and renumbering it as Fig 1

In my opinion this paper is suitable for publication in “Plants” after minor revision

Comments on the Quality of English Language

I suggest a careful revision of this paper; in different parts of the manuscript the sentences are not clear or contain typos
